# Marine Natural Products Rescuing the Eye: A Narrative Review

**DOI:** 10.3390/md22040155

**Published:** 2024-03-28

**Authors:** Filippo Lixi, Livio Vitiello, Giuseppe Giannaccare

**Affiliations:** 1Eye Clinic, Department of Surgical Sciences, University of Cagliari, 09124 Cagliari, Italy; f.lixi1@studenti.unica.it; 2Eye Unit, “Luigi Curto” Hospital, Azienda Sanitaria Locale Salerno, 84035 Polla, Italy; livio.vitiello@gmail.com

**Keywords:** eye, marine natural products, AMD, glaucoma, dry eye disease

## Abstract

Different degrees of visual impairment lead to a decrease in patient wellbeing, which has an adverse effect on many facets of social and professional life. Eye disorders can affect several parts of the eye, most notably the retina and the cornea, and the impacted areas might share a common form of cellular damage or dysfunction (such as inflammation, oxidative stress and neuronal degeneration). Considering that marine organisms inhabit a broad variety of marine habitats, they display a great degree of chemical diversity. As a result, molecules with a marine origin are receiving more and more attention in the hopes of developing novel therapeutic approaches. For instance, fucoxanthin has been demonstrated to be effective in protecting the retina against photo-induced damage, while largazole, astaxanthin and spirulina have all shown antioxidant, anti-inflammatory and antiapoptotic activities that can be useful for the management of several ocular diseases, such as age-related macular degeneration and ocular surface disorders. The aim of this review is to analyze the scientific literature relating to the therapeutic effects on the eye of the main natural marine products, focusing on their mechanism of action and potential clinical uses for the management of ocular diseases.

## 1. Introduction

Among the most common causes of blindness in the world, there are multifactorial conditions like glaucoma, age-related macular degeneration (AMD) and diabetic retinopathy (DR), which are linked to the influences of multiple genes as well as environmental factors [1,2]. Although the underlying molecular and cellular mechanisms of these disorders are currently poorly understood, the common features of these diseases include the degeneration and death of retinal pigment epithelium (RPE) and/or photoreceptor cells, as well as retinal ganglion cells (RGCs). The hallmarks of the retinal cell death process, however, are frequently altered processes that worsen the course of the illness, such as inflammation, mitochondrial dysfunction and microglia activation. The development of gene-/mutation-specific therapy techniques that can be applied to a large proportion of patients is hindered by the high genetic variability and complex character of various eye diseases. These factors have led to an increased interest in complementary and alternative therapies that work on common pathways underlying ocular damage to slow the course of these pathologies using mutation-independent treatment techniques [3].

Recently, the use of marine resources in the management and prevention of ocular disorders has been strongly in focus [4]. At least half of the bioactive compounds found in the marine environment have prospective uses related to human health [4]. These molecules can be employed in nutritional, cosmetic, medicinal and other biotechnological products. Numerous studies have reported the neuroprotective effects of marine natural products in the context of neurodegenerative diseases [5]. Additionally, a number of studies have highlighted the potential role that marine compounds may play in the prevention or delaying of various ocular diseases through their anti-inflammatory, antioxidant, antiangiogenic/vasoprotective and cytoprotective activities [4].

The purpose of this review is to examine the scientific literature about the main natural marine products’ therapeutic effects on the eye, with an emphasis on their mechanism of action and possible clinical applications for the treatment and management of ocular disorders.

## 2. Materials and Methods

For our literature search, we used the PubMed medical database. The terms “eye”, “retina”, “cornea” and “lens” were combined with a number of “text words” associated with natural marine compounds that have shown potential ocular-positive and beneficial effects (e.g., “fucoxanthin”, “astaxanthin”, “largazole”, “spirulina”, “oxasqualenoids”, “sesquiterpenes”, “fucoidan”, “diphlorethohydroxycarmalol”). Text keywords were selected from the most recent literature and relevant bibliographies. The search was carried out with an end date of February 2024 and a start publishing date of January 1990. Although there were no language restrictions on the search, only English-language articles and reviews were examined. Furthermore, manual searches were conducted within the first findings to uncover other bibliographic entries.

## 3. Results

### 3.1. Fucoxanthin

Fucoxanthin is an orange carotenoid with light-absorbing properties, abundantly synthesized in nature by different varieties of brown seaweeds such as *Hijikia fusiformis*, *Laminaria japonica* and *Sargassum fulvellum*. Its antioxidant, anti-inflammatory, anti-tumor, antimicrobial and lifespan-prolonging effects have been extensively described in different medical fields [6,7,8]. In addition, fucoxanthin administration was investigated in order to manage sight disorders [9]. The retina is the nervous part of the eye capable of transforming light rays into electrical impulses. However, light generates oxidative stress and reactive oxygen species (ROS), which are responsible for the light-induced retinal damage characterized by photoreceptor loss. In this regard, Liu et al., on in vitro and in vivo models of visible light-induced retinal damage, showed that fucoxanthin use inhibited the overexpression of vascular endothelial growth factor (VEGF), improving phagocytic function and reducing ROS. In vivo tests also confirmed the effect of fucoxanthin in protecting the retina against photo-induced damage, showing a stronger action than lutein [10]. Different mechanisms were proposed to explain the function of this molecule. The good ability to absorb blue light, which is considered the principal factor responsible for retinal injuries owing to its ability to penetrate tissues, could be the reason for the protective effect of fucoxanthin on retinal cells. Furthermore, this orange carotenoid has particular structural features that offer an exceptional ability to bond radicals and improve its bioactivity [10]. Age-related macular degeneration (AMD) is the leading cause of vision impairment among elderly populations in developed countries. It occurs due to degeneration of photoreceptors and RPE secondary to several risk factors including age, obesity, cigarette smoking and light exposure, resulting in high oxidative stress and inflammation. In a sodium-iodate-induced AMD animal model, fucoxanthin exhibited a reduction in photoreceptor death and protective properties on RPE cells subject to oxidative damage [11]. Similarly, in vitro, after peroxide-induced stress, fucoxanthin presented protective properties on a human RPE cell line, ARPE-19 cells, displaying an inhibition of cell death, reduction in ROS, reduction in microvilli disruption and amyloid-beta deposition [11]. Uveitis are inflammatory conditions involving the vascular tunica of the eye. The anti-inflammatory properties of fucoxanthin were assessed in a study on endotoxin-induced uveitis (EIU) in rats, where it was demonstrated to lower levels of nitric oxide, prostaglandin-E2 and tumor necrosis factor (TNF)-α in the aqueous humor in a similar way to prednisolone [12]. The preventive administration of fucoxanthin may also protect the cornea from ultraviolet B-induced damage. Indeed, in mice, it preserved the corneal surface and reduced inflammatory cellular infiltration and cytokines (TNF-α, VEGF) [13]. Furthermore, treatment with fucoxanthin has displayed the potential to both shield the eyes from ultraviolet B-induced photokeratitis by limiting corneal denervation and to alleviate inflammatory neuropathic pain through the expression of the antioxidant nuclear factor erythroid 2-related factor 2 (Nrf2) and the activation of neural cells of the trigeminal ganglia [14] (Table 1).

### 3.2. Astaxanthin

Astaxanthin is a red pigmented xanthophyll, common in marine environments, where it is bioproduced by marine microorganism, such as *Haematococcus pluvialis*, *Agrobacterium aurantiacum*, *Chlorella zofingiensis* and *Xanthophyllomyces dendrorhous*. It has demonstrated various positive effects, including its anti-cancer, anti-apoptotic, anti-inflammatory, antioxidant and neuroprotective properties [38,39]. Concerning eye wellness, beneficial activities in various ocular conditions such as AMD, diabetic retinopathy, glaucoma, dry eye, uveitis and cataract were reported [40]. Astaxanthin, thanks to its strong antioxidant effect and its scavenging activity, has demonstrated the capacity to reduce ROS production in vitro, as well as to reduce retinal ischemic cell death and light-induced damage in vivo, as assessed through an electroretinogram and histologic evaluation [41,42]. In a human study, Parisi et al. randomly divided up a group of 27 people with non-advanced AMD and tested dietary supplementation over one year with different substances including astaxanthin. At the end of follow-up, the treated group presented an improvement of retinal function, as evaluated by an electroretinogram, in comparison with the control group [43]. In addition, the Carmis study group in a two-year, multicentre, prospective study reported a significant positive effect on visual acuity, contrast sensitivity and subjective visual function in patients with AMD treated with an oral supplementation of lutein, zeaxanthin and astaxanthin [44]. In exudative neovascular AMD, new capillaries arise from the choroid through Bruch’s membrane and enter the retina. In an animal study, the role of astaxanthin was investigated in the management of neovascular AMD. Choroidal neovascularization (CNV), induced by laser photocoagulation, was suppressed by astaxanthin treatment. Indeed, CNV develops in the presence of inflammation and VEGF upregulation, which were significantly reduced by astaxanthin administration, which suppressed the nuclear factor-kappaB (NF-κB) pathway and consequently other inflammatory and angiogenic molecules such as VEGF, VEGFR-1, VEGFR-2, IL-6, ICAM-1 and MCP-1 [45]. Diabetic retinopathy (DR), the leading cause of vision loss in the working-age population, is characterized by vascular and cellular damage in the eye posterior pole due to the activation of several molecular mechanisms including the polyol, advanced end glycation (AGE) product, hexosamine, protein kinase and tissue renin–angiotensin system (RAS) pathways [46]. In a study on rats with induced diabetes, astaxanthin proved to inhibit the activity of transcription factor NF-κB, to reduce oxidative stress and inflammatory mediators and to increase the levels of antioxidant enzymes like heme oxygenase-1 and peroxiredoxin [47]. In another report, Benlarbi-Ben Khedher et al. discovered, either in vitro or in vivo after astaxanthin treatment, a reduced activity of aldolase reductase, which is the main enzyme of the polyol pathway, highly implicated in DR microvascular alterations [48] (Table 2). Glaucoma is a chronic, progressive, optic neuropathy characterized by nerve fiber layer and visual field loss. Although the pathogenesis is still not completely understood, the optic nerve damage occurs as a result of mechanical compression due to high intraocular pressure (IOP) or secondary to vascular alterations. Currently, the therapy for glaucomatous patients is based on IOP-lowering strategies including drugs, lasers and surgery if conservative treatments’ results are unsuccessful [49,50]. However, a possible progression of the disease even with a well-controlled IOP has led to suggestions to explore other targets for slowing glaucoma evolution, such as oxidative stress [51]. Indeed, as previously reported, astaxanthin, beside its well-documented anti-inflammatory and antioxidant properties, has proved to have a good neuroprotective effect. In this regard, Cort et al. studied the neuroprotective action of astaxanthin in mice with elevated IOPs. In the study group, astaxanthin reduced apoptotic cells’ percentage and oxidative markers. In addition, it exhibited a neuroprotective effect when evaluated through electrophysiological measurements of visual evoked potentials, which were altered by IOP elevation and subsequently restored after astaxanthin administration [52]. The neuroprotective action of this xanthophyll was also studied in a rat model of nonarteritic anterior ischemic optic neuropathy. Mice received astaxanthin supplementation either before or after inducing ischemic stress in the retina. In both the pre- and post-treatment groups, there were notable increases in visual function and RGC densities. Additionally, in both groups, oxidative and inflammatory markers were lowered [53]. The anti-inflammatory action of astaxanthin was investigated in an endotoxin-induced uveitis animal model [15,16]. Ohgami and colleagues reported that astaxanthin inhibited the progression of endotoxin-induced uveitis in a dose-dependent manner, exhibiting a similar efficacy to prednisolone. Additionally, astaxanthin reduced the nitric oxide, prostaglandin-E2 and TNF-α production in vitro [15]. Suzuki and co-authors had similar results as well, reporting a reduction in the NF-kB activation pathway and inflammatory cytokine levels in the aqueous humor of rats [16]. Cataracts are still one of the main causes of blindness and surgical removal remains the elective therapy for restoring vision. Cataract formation is an age-related process in which lens proteins change their composition, resulting in the formation of opacities. This process can be accelerated by different conditions such as hyperglicemia, steroid use and trauma, resulting in heightened oxidative stress. The use of astaxanthin was explored in animal models of metabolic cataracts, showing promising results [17,18]. Indeed, in a chick embryo model of steroid-induced cataracts, astaxanthin administration reduced lens opacification and restored glutathione levels [17]. Similarly, Yang and co-authors reported in a diabetic mouse model that astaxanthin decreased the severity of cataracts and inhibited oxidative stress by lowering the levels of AGE products and lipid peroxide/malondialdehyde in lens tissue [18]. Dry eye disease (DED) is a multifactorial disorder characterized by a loss of homeostasis of the tear film associated with ocular symptoms, in which a significant rise in oxidative stress and inflammation plays a pivotal role in inducing the disease [19]. Shimokawa and collegues confirmed the relationship between ROS and age-related markers in an in vitro dry eye model and demonstrated the potential of high-affinity liposomal astaxanthin for the treatment of dry eye. Furthermore, the use of astaxanthin and other nutraceutical products was evaluated in a prospective randomized study, in which an antioxidant compound containing anthocyanosides, astaxanthin, vitamins A, C and E and other herbal extracts was compared with a placebo in patients with DED. The results showed improved tear film breakup time (BUT) scores and Schirmer’s test scores and lowered ROS levels in the study group, confirming that oral antioxidant supplementations may increase tear production and improve tear film stability [20]. Asthenopia, commonly referred to as eye strain, is a frequent condition characterized by unspecific symptoms (discomfort, tearing, blurred vision, sensitivity to light, headache) which may be exacerbated by computer and display use. Astaxanthin’s role in relieving eye fatigue has been extensively studied in the last years [21,22,23,24,25,26]. Nagaki et al. demonstrated in a randomized clinical trial that astaxanthin supplementation reduced eye muscle strain and improved accommodation amplitude [21]. Improvement of uncorrected visual acuity, accommodation and subjective symptoms related to visual display or terminal use were reported in healthy young adults who received oral astaxanthin either alone or in association with other substances [22,23]. Even in middle-aged and elderly people, astaxanthin improved accommodation and subjective symptoms of asthenopia such as a stiff neck or shoulders and blurred vision [24,25,26]. The mechanism for explaining the activity of astaxanthin in managing asthenopia is not completely understood; however, it may be related to the strong antioxidative effect and to the increased blood flow in the ciliary muscles, which improve the accommodative function [23,26] (Table 1).

### 3.3. Largazole

Largazole is a natural product isolated from marine cyanobacteria of the genus *Symploca*, which has shown promise in various biological activities thanks to its capacity to selectively inhibit class I histone deacetylases (HDACs). HDACs control gene expression by modifying the structure of chromatin, which is the complex of DNA and proteins found in the nucleus of cells. Through its inhibition, largazole downregulates the expression of pro-angiogenetic factors and upregulates anti-angiogenetic mediators [27,54]. Therefore, largazole has been explored in different eye disorders for its anti-inflammatory and anti-angiogenetic properties [27,54]. Zhou et al. evaluated the effect of largazole on inflammatory corneal angiogenesis in a mouse model of alkali-induced corneal neovascularization. Topical application of largazole resulted in attenuation of corneal neovascularization formation by reducing the levels of angiogenic factors (VEGF, b-FGF, TGFβ1 and EGF) and by increasing anti-angiogenetic factors (Thrombospondin-1 (Tsp-1), Tsp-2 and ADAMTS-1) in the injured corneas. In addition, largazole demonstrated inhibition of the expression of pro-angiogenic factors, migration, proliferation and tube formation by human microvascular endothelial cells in vitro [27] (Table 1). Similarly, Qiu and co-authors confirmed the anti-angiogenetic properties of largazole in a study on neovascularization in the posterior eye segment. Specifically, they reported largazole inhibition of retinal vascular endothelial cell viability and proliferation in vitro. Moreover, in ex vivo models, largazole reduced the vessel outgrowth from choroidal explants in a choroid sprouting assay and demonstrated a cooperative effect with aflibercept, an FDA-approved anti-VEGF drug currently used in clinical practice [54] (Table 2).

### 3.4. Spirulina

Spirulina is a biomass constituted by *Arthrospira platensis*, *Arthrospira fusiformis* and *Arthrospira maxima*, which all belong to the *Cyanobacteria* species. This blue-green alga can be found in various marine environments, particularly in mineral-rich water, and it is considered a superfood thanks to its high content of nutrients (e.g., proteins, amino acids, vitamins, fatty acids, pigments, minerals and phytonutrients) [67]. Furthermore, spirulina is employed in medicine for treating various health conditions, owing to its different properties such as antioxidant, immuno-modulatory, anti-inflammatory, antimicrobic, antitumor, antiallergic and antidiabetic activities [68,69]. Spirulina effect’s was previously investigated in different ophthalmic disorders [70]. In a corneal alkali burn model, topical application of spirulina significantly inhibited corneal neovascularization. Corneas treated with spirulina exhibited lowered levels of inflammatory and pro-angiogenic factors including CD31, TNF-α, matrix metalloproteinase-2, matrix metalloproteinase-9 and VEGF. In human vascular endothelial cells, spirulina significantly inhibited proliferation, migration and tube formation in a dose-dependent manner [28]. Additionally, a polysaccharide extract from Spirulina platensis was demonstrated to inhibit alkali-burn-induced inflammation and corneal neovascularization more effectively than amnion membrane extract [29] (Table 1). Spirulina also exhibited a good radioprotective effect on the lacrimal glands after radioactive iodine therapy in an animal model. In fact, after radiotherapy, spirulina administration showed a reduction in inflammatory factors and of the total oxidant status during histopathological and cytopathological analysis [71]. Furthermore, spirulina demonstrated good protection from light-induced retinal damage secondary to oxidative stress and inflammation. In an animal study, mice that belonged to the spirulina-supplemented group, after light exposure, exhibited a reduction in thinning of the photoreceptor layer and outer segments, photoreceptor apoptosis and retinal ROS levels. Two possible pathways for suppressing ROS were proposed: a direct or indirect scavenging activity by spirulina or by its component phycocyanin, which is a water-soluble, radical-scavenger pigmented protein, and, alternatively, an induction of endogenous antioxidative enzymes (e.g., nuclear Nrf2) [55]. Similarly, Cho et al. demonstrated that spirulina had protective effects on blue-light-induced cell death by regulating ROS production through phycocyanin, the main active compound of spirulina maxima. Particularly, in vitro, spirulina exhibited anti-inflammatory and anti-apoptotic effects through downregulating the transcription factor NF-κB, inflammation and apoptosis in RPE cells. Moreover, in vivo, spirulina restored the thicknesses of several retinal layers remarkably reduced by light exposure, thus proving to be a potential therapeutic agent to prevent AMD [56] (Table 2).

### 3.5. Oxasqualenoids

The red alga *Laurencia viridis* produces a huge variety of natural biologically active metabolites that exhibit cytotoxic, antiviral and anti-inflammatory activity [30]. *Acanthamoeba* spp. are free-living amoebae, ubiquitarians in nature, which live as opportunistic pathogens in humans and animals. Acanthamoeba keratitis (AK) is the main ocular manifestation caused by *Acanthamoeba* spp. and represents an aggressive, sight-threatening condition. In cases of AK, the parasite adheres to the corneal surface by binding to glycoproteins on the epithelium, successively delivers enzymes and toxins (proteases, phospholipases, cytopathic and collagenolytic factors) and invades the stroma, causing an inflammatory response that leads to corneal cell death and melting [72]. Lorenzo-Morales and colleagues tested different oxasqualenoids extracted by the red seaweed Laurencia viridis (dehydrothyrsiferol, thyrsiferol, iubol, 22-hydroxydehydrothyrsiferol, 1,2-dehydropseudodehydrothyrsiferol, saiyacenols A and B, 28-hydroxysaiyacenol A and B, nivariol A and the truncated metabolite adejene B) against *Acanthamoeba castellanii Neff*. All tested oxasqualenoid molecules demonstrated positive activity against *Acanthamoeba castellanii Neff*, proving to be possible candidates for the development of new amoebicidal therapies [30] (Table 1).

### 3.6. Sesquiterpenes

Terpenes are natural compounds with several biological activities including antimicrobial, antifungal, antiviral, anti-inflammatory and antitumoral. Terpenes exhibit structural diversity and are classified based on the number of isopentenyl pyrophosphate units they contain into hemiterpene (1 unit), monoterpene (2 units), sesquiterpene (3 units); diterpene (4 units), triterpene (6 units), tetraterpene (8 units) and polyterpene (over 8 units) [73]. Concerning ocular diseases, García-Davis et al. studied the antiparasitic activity of different sesquiterpenes isolated from the red alga Laurencia johnstonii. In particular, brominated sesquiterpenes α-bromocuparane and α-isobromocuparane demonstrated amoebicidal activity against Acanthamoeba castellanii Neff. In addition, other inactive sesquiterpenes, after chemical transformation, showed increased antiamoeboid activity [31] (Table 1).

### 3.7. Fucoidan

Fucoidan is a long-chain sulfated polysaccharide found in various species of brown marine algae such as *Fucus vesiculosus*, *Cladosiphon okamuranus*, *Laminaria japonica*, *Sargassum horneri* and *Undaria pinnatifida*. Thanks to different positive and therapeutic properties (anti-inflammatory, antioxidant, antiangiogenetic, anti-diabetic, anticancer), fucoidan has been evaluated for uses in the healthcare sector and especially in ophthalmology, where it has demonstrated positive effects in managing conditions such as AMD, diabetic retinopathy, dry eye disease and proliferative vitreoretinopathy (PVR) [9,74]. Fucoidan was shown to inhibit diabetic retinal neovascularization through the downregulation of hypoxia-inducible factor (HIF)-1α/VEGF signaling [57]. In addition, Li and co-authors reported that fucoidan, by reducing ROS formation through the antioxidative Ca^2+^-dependent ERK signaling pathway, protected ARPE-19 retinal cells from high glucose-induced cell death and apoptosis [58]. Dithmer et al. investigated the use of fucoidan for the treatment of exudative AMD in ARPE-19 and primary porcine RPE cells, displaying a good safety profile and a reduction in VEGF secretion on RPE/choroid explants and ARPE-19 cells [59]. Tear hyposecretion is one of the main causes of dry eye. In affected patients, the hyperosmolar environment causes the beginning of a vicious circle characterized by inflammation and cellular loss in the ocular surface [75]. According to this, fucoidan administration was studied for the prevention of dry eye disease in corneal epithelial cells and in rats with induced dry eye disease after excision of their lacrimal gland. It was found that fucoidan suppressed apoptosis and the expression of apoptosis-related proteins in human corneal epithelial cells under hyperosmotic conditions. Moreover, fucoidan reduced tear hyposecretion and corneal irregularity in the lacrimal-gland-excised rats [32] (Table 1). A challenging sight-endangering complication of retinal detachment surgery is PVR. The latter is characterized by the formation of an epi-/sub-retinal membrane and traction of the reattached retina. The RPE cells are considered to play an important role in the pathogenesis of PVR by losing their epithelial properties and transforming into mesenchymal cells capable of migrating and proliferating [60]. Zhang et al. tested fucoidan on the epithelial–mesenchymal transition of ARPE-19 cells, finding that this marine product was able to reverse this transition by inhibiting TGF-β1 and increasing the expression of α-smooth muscle actin (α-SMA) and fibronectin. Furthermore, fucoidan proved to block the progression of experimental PVR in rabbit eyes, suppressing the formation of α-SMA-positive epiretinal membranes [61] (Table 2).

### 3.8. Diphlorethohydroxycarmalol

Diphlorethohydroxycarmalol (DPHC) is a phlorotannin compound isolated from the brown alga *Ishige okamure Yendo*. This phenolic agent has various positive properties including antioxidant activity and ROS scavenging [76]. Park et al. suggested the possible role of DPHC in preventing AMD. Indeed, DPHC protected ARPE-19 cells against H_2_O_2_-induced DNA damage and apoptosis by scavenging ROS and thus suppressing the mitochondrial-dependent apoptosis pathway [62] (Table 2).

### 3.9. Heparin-like Compound

Heparin is a sulfated polysaccharide well known for its anticoagulant behaviour. Beyond that, it has also demonstrated anti-inflammatory and anti-angiogenetic activities. However, its clinical use as anti-angiogenetic agent is limited by its strong anticoagulant activity and possible related haemorrhagic complications [77,78]. Recently, a heparinoid isolated from marine shrimp *Litopenaeus vannamei* presenting negligible anticoagulant and haemorrhagic potential was shown to be capable of reducing acute inflammatory processes in an animal model [79]. Therefore, Dreyfuss and colleagues evaluated the same heparin-like compound to assess its anti-angiogenic action and its possible use for treating neovascular AMD. This heparinoid was shown to have anti-angiogenetic activity in laser-induced CNV in mice and in vitro models. In particular, this compound decreased growth factors (TGF-β1, FGF-2, EGF and VEGF), reduced the CNV area and blocked endothelial cell proliferation without showing a cytotoxic effect [63].

### 3.10. Homotaurine

Homotaurine (also known as tramiprosate) is a natural amino sulfonate compound derived from marine red algae (*Hypnea boergesenii*, *Gracilaria corticate* and *Gracilaria pygmae*) [80]. This substance is known to have neuromodulatory properties and the ability to improve nerve impulse transmission in the central nervous system [81]. In both in vitro and in vivo experimental settings, homotaurine can interfere with a variety of biological processes and its cytoprotective, neuroprotective and neurotropic properties have already been noted. Additionally, it is a GABA analog, which means that it has strong agonistic effects on GABA receptors, preferring GABA type A receptors. This feature supports its analgesic and anti-nociceptive effects, which are likely mediated by cholinergic and opioid pathways [82]. Furthermore, homotaurine may offer some resistance against cellular damage, particularly against free-radical-induced oxidative damage to DNA [82], and it also has the capacity to stop the development of β-amyloid plaques, which are connected to neuronal cell death and other neurodegenerative processes in the central nervous system [81]. In fact, previous research has highlighted the possibility that amyloid plays a role in the activation of RGCs’ apoptosis in models of experimental glaucoma [64], suggesting that homotaurine may prove to be a useful medication for glaucoma. According to a recent study, citicoline and homotaurine together have strong synergistic neuroprotective effects in cultured retinal cells, which lessen the proapoptotic effects of exposure to glutamate and high glucose [65]. In addition, another randomized, controlled, multicentric trial showed that daily oral administration of a fixed combination of citicoline and homotaurine for four months improved inner retinal cell function. Moreover, the visual field and the perception of quality of life were both positively affected by this improvement, which happened independently of IOP decrease [66] (Table 2). For this reason, homotaurine could be considered a potential neuroprotective compound that could reduce RGC loss thanks to its cytoprotective and neurotropic properties, helping in the management of glaucoma. However, future studies are needed to better understand its potential role in clinical practice.

### 3.11. Quinoid Pigments

Quinoid pigments are organic phenolic compounds isolated from the shells and spines of sea urchins. These have been extensively studied for their wide potential utilities and particurly for their medical applications [83]. Moreover, thanks to their anti-oxidant, anti-bacterial, anti-viral, anti-inflammatory and anti-allergic properties, quinoids have been utilized in managing ocular disorders. Pozharitskaya and collegues evaluated the effect of an extract of pigments from the green sea urchin Strongylocentrotus droebachiensis predominantly containing polyhydroxy-1,4-naphthoquinone. In rabbits, the extract showed a therapeutic effect, without irritation, on induced allergic conjunctivitis, displaying a more effective action in comparison with olopatadine, the reference drug for ocular allergies [33]. Echinochrome A, one of the most abundant quinonoid pigments in sea urchins, proved to be effective in treating traumatic hemophthalmia in rabbits. Moreover, the administration of Echinochrome A enhanced corneal epithelialization, decreased corneal defects, reduced the risk of corneal perforation and alleviated eye inflammation [34] (Table 1). Thus, Histochrome, a drug preparation containing sodium and echinochrome A, was approved in Russia for the treatment of several eye conditions including glaucoma, intraocular haemorrhages, degenerative retinal and corneal diseases and vascular and inflammatory ocular disorders [35]. Clinical studies have demonstrated that Histochrome effectively improves visual acuity and reduces hemorrhage and inflammation [35,36]. In particular, it has demonstrated efficacy in resolving hyphema, retinal hemorrhages and diabetic retinopathy. Additionally, Histochrome has exhibited retinoprotective properties and enhanced blood flow in the posterior pole of the eye. Positive results were also obtained when keratitis and corneal dystrophy were treated with Histochrome [37]. In addition, in patients with primary open-angle glaucoma (POAG) undergoing trabeculectomy, combining Histochrome with magnetotherapy led to improved visual function and stabilized glaucoma progression during the early postoperative period [35] (Table 2).

## 4. Discussion

From a sociological standpoint, eye diseases can significantly impair patients’ quality of life and place a financial strain on the healthcare system.

The majority of ocular pathologies that result in blindness are age-related, and the fact that the world’s population is growing older should be seriously considered. These observations underline the growing need to find safe and efficient treatment approaches that might improve patients’ quality of life and halt the course of ocular diseases, regardless of their etiology. 

Throughout the past century, an increasing number of marine natural products that have the capacity to control physiological processes have been identified and extracted from plants, animals and microorganisms, showing a great deal of promise for application in medicine [84]. Due to their good safety and efficacy profiles demonstrated in clinical trials, their applications have supported important advancements in the medical field. In the future, it is likely that patients’ quality of life and therapy outcomes will be improved with the discovery of novel, potent and low-toxicity therapeutic agents.

Considering the great biodiversity, genetic distinctiveness and fierce rivalry among marine creatures for survival in their ecosystem, which is frequently mirrored in their chemistry and bioactivity, marine natural products have long been recognized for their structural variety [85].

Starting from all these considerations, this review has focused on the main marine natural products that have already demonstrated encouraging therapeutic effects in some ocular pathologies.

Concerning the anterior segment of the eye (Table 1), several marine molecules such as fucoxanthin, astaxanthin, largazole and spirulina have demonstrated potential therapeutic effects in case of uveitis and corneal neovascularization, reducing the levels of inflammatory cytokines and pro-angiogenetic factors and upregulating the levels of anti-angiogenetic factors. Furthermore, some marine compounds such as oxasqualenoids and sesquiterpenes have shown promising antimicrobial effects against Acanthamoeba species, responsible for a very sight-threatening form of keratitis.

In addition, fucoidan, spirulina and astaxanthin seem to be promising agents for DED management, thanks to their antioxidant activity.

Regarding the posterior segment of the eye (Table 2), all the analyzed natural marine products show therapeutic effects thanks to their antioxidant, anti-inflammatory and anti-angiogenetic activities, in particular preventing photo-induced damage to retinal cells and helping to reduce the harmful effects of eye conditions such as AMD and diabetic retinopathy. In addition, astaxanthin and homotaurine seem to have neuroprotective effects against glaucoma, reducing RGC loss, while fucoidan is able to inhibit PVR onset after retinal detachment surgery in animal models. Moreover, all these molecules show an excellent safety profile, guaranteeing their possible use in clinical practice [4].

In Table 3, the main clinical studies concerning the natural marine molecules with potential therapeutic effects on the eye are summarized.

## 5. Conclusions

In conclusion, it should be noted that marine organisms and their natural molecules may provide a highly attractive chemical pool for the discovery of novel marine microbial opsins and compounds with unique properties beneficial for ocular diseases, as well as pharmacologically active substances with new and innovative structures and activities, as has already happened for other natural treatments [86]. Future research into the deep sea and other extreme marine settings may provide marine species that may be studied and characterized to provide new treatment approaches for blindness and other ocular and human diseases whose progression involves comparable dysregulated molecular processes.

## Figures and Tables

**Table 1 marinedrugs-22-00155-t001:** Summary of the main effects of the discussed marine natural products on the anterior segment of the eye.

Marine Compound	Sources	Effects	Ref.
Fucoxanthin	*Hijikia fusiformis*, *Laminaria japonica*, *Sargassum fulvellum*	Lower levels of nitric oxide, prostaglandin-E2 and tumor necrosis factor α in the aqueous humor in uveitis Protect the cornea from ultraviolet B-induced damage, reducing inflammatory cellular infiltration and cytokines	[12,13,14]
Astaxanthin	*Haematococcus pluvialis*, *Agrobacterium aurantiacum*, *Chlorella zofingiensis*, *Xanthophyllomyces dendrorhous*	Lower levels of nitric oxide, prostaglandin-E2 and tumor necrosis factor α in the aqueous humor in uveitis Reduce lens opacification and restore glutathione levels in animal model of steroid-induced cataracts Improve tear film breakup time and Schirmer’s test scores, along with lowering reactive oxygen species levels, in dry eye disease Relieve eye fatigue in asthenopia	[15,16,17,18,19,20,21,22,23,24,25,26]
Largazole	*Cyanobacterium Symploca* sp.	Downregulate the expression of pro-angiogenetic factors and upregulate anti-angiogenetic mediators in corneal neovascularization	[27]
Spirulina	*Arthrospira platensis*, *Arthrospira fusiformis*, *Arthrospira maxima*	Inhibit corneal neovascularization in a corneal alkali burn model, lowering levels of inflammatory and proangiogenic factors Exhibit a good radioprotective effect on the lacrimal glands after radioactive iodine therapy in an animal model	[28,29]
Oxasqualenoids	*Laurencia viridis*	Potential amoebicidal effect in Acanthamoeba keratitis	[30]
Sesquiterpenes	*Laurencia johnstonii*	Potential amoebicidal effect in Acanthamoeba keratitis	[31]
Fucoidan	*Fucus vesiculosus*, *Cladosiphon okamuranus*, *Laminaria japonica, Sargassum horneri*, *Undaria pinnatifida*	Suppress apoptosis and the expression of apoptosis-related proteins in human corneal epithelial cells under hyperosmotic conditions	[32]
Quinoid pigments	*Strongylocentrotus droebachiensis*	Therapeutic effect on induced model of allergic conjunctivitis Promote epithelialization of the cornea, as well as reduce ocular inflammation and the risk of corneal perforation	[33,34,35,36,37]

**Table 2 marinedrugs-22-00155-t002:** Summary of the main effects of the discussed marine natural products on the posterior segment of the eye.

Active Compound	Sources	Effects	Ref.
Fucoxanthin	*Hijikia fusiformis*, *Laminaria japonica*, *Sargassum fulvellum*	Inhibit the overexpression of vascular endothelial growth factor Improve phagocytic function Reduce reactive oxygen species Protect retina against photo-induced damage Reduce photoreceptors death in age-related macular degeneration animal models Protect retinal pigment epithelium cells	[10,11]
Astaxanthin	*Haematococcus pluvialis*, *Agrobacterium aurantiacum*, *Chlorella zofingiensis*, *Xanthophyllomyces dendrorhous*	Reduce reactive oxygen species production, retinal ischemic cell death and light-induced damage Downregulate several inflammatory and angiogenic molecules in age-related macular degeneration Reduce oxidative stress and inflammatory mediators, increasing levels of antioxidant enzymes in diabetic retinopathy Exhibit neuroprotective effects in glaucoma, reducing apoptotic cells’ percentage and oxidative markers	[41,42,43,44,45,46,47,48]
Largazole	*Cyanobacterium Symploca* sp.	Downregulate the expression of pro-angiogenetic factors and upregulate anti-angiogenetic mediators in choroidal neovascularization	[54]
Spirulina	*Arthrospira platensis*, *Arthrospira fusiformis*, *Arthrospira maxima*	Demonstrate good protection from light-induced retinal damage secondary to oxidative stress and inflammation, reducing thinning of the photoreceptor layer and outer segments, photoreceptor apoptosis and retinal reactive oxygen species levels	[55,56]
Fucoidan	*Fucus vesiculosus*, *Cladosiphon okamuranus*, *Laminaria japonica*, *Sargassum horneri*, *Undaria pinnatifida*	Inhibit retinal neovascularization through the downregulation of hypoxia-inducible factor-1α/vascular endothelial growth factor signaling in diabetic retinopathy Reduce vascular endothelial growth factor production in exudative age-related macular degeneration animal model Inhibit proliferative vitreoretinopathy onset after retinal detachment surgery in animal models	[57,58,59,60,61]
Diphlorethohydroxycarmalol	*Ishige okamure Yendo*	Potential role in preventing age-related macular degeneration by scavenging reactive oxygen species	[62]
Heparin-like compound	*Litopenaeus vannamei*	Reduce the choroidal neovascularization area and block endothelial cell proliferation without showing a cytotoxic effect in mice and in vitro models	[63]
Homotaurine	*Hypnea boergesenii*, *Gracilaria corticate*, *Gracilaria pygmae*	Stop the development of β-amyloid plaques, reducing the retinal ganglion cells’ apoptosis	[64,65,66]
Quinoid pigment	*Strongylocentrotus droebachiensis*	Ehnance blood flow in the posterior pole of the eye, exhibit retinoprotective properties and reduce glaucoma progression	[35,36]

**Table 3 marinedrugs-22-00155-t003:** Summary of the main clinical studies investigating marine natural products’ effects on the eye.

Author (Year)	Molecule	Type of Study	Outcomes	Ref.
Giannaccare et al. (2020)	Astaxanthin	Review	Efficacy in the prevention and treatment of several ocular diseases, ranging from the anterior to the posterior pole of the eye (retinal diseases, ocular surface disorders, uveitis, cataracts and asthenopia)	[40]
Parisi et al. (2008)	Astaxanthin and other compounds	Human study on 27 patients, 15 patients with oral supplementation of vitamin C (180 mg), vitamin E (30 mg), zinc (22.5 mg), copper (1 mg), lutein (10 mg), zeaxanthin (1 mg) and astaxanthin (4 mg) daily for 12 months, whereas 12 patients had no dietary supplementation	Improvement of central retinal function in the study group	[43]
Huang et al. (2016)	Astaxanthin and other antioxidant compounds	Human study on 43 patients, with 20 in treatment and 23 in placebo group. An antioxidant supplement (containing anthocyanosides, astaxanthin 20 mg, vitamins A, C and E and several herbal extracts, including Cassiae semen and Ophiopogonis japonicus) was compared with placebo on patients with dry eye	No significant differences in dry eye symptoms, serum anti-SSA and anti-SSB, visual acuity, intraocular pressure or fluorescein corneal staining between the groups. Tear film breakup time scores and Schirmer’s test significantly improved in the treatment group	[20]
Kizawa et al. (2021)	Astaxanthin, anthocyanin and lutein	Human study on 44 patients, where two active (astaxanthin 60 mg/capsule, anthocyanin 100 mg/capsule, lutein 12.5 mg/capsule) or placebo capsules were taken once daily for 6 weeks	Significant improvement in the study group in the percentage of pupillary response. Moreover, the study group showed a significant improvement in subjective eye fatigue and accomodation	[23]
Sekikawa et al. (2022)	Astaxanthin (9 mg/day) or placebo	Human study on 60 patients. Participants received a diet containing astaxanthin (9 mg/day) (*n* = 30) or placebo (*n* = 30) for 6 weeks	In participants aged ≥40 years, corrected visual acuity of the dominant eye after visual display terminal work at 6 weeks after intake demonstrated a higher protective effect of astaxanthin in the study group vs. the control group. In participants aged <40 years, no significant difference was seen between the two groups. No significant difference was found in functional visual acuity or pupil constriction rate between the two groups	[24]
Rossi et al. (2022)	Homotaurine 50 mg and Citicoline 500 mg	Human study on 57 patients with primary open-angle glaucoma with controlled IOP, 26 in group A and 31 in group B (group A: topical therapy + CIT/HOMO for 4 months, 2 months of wash out, 4 months of topical therapy alone; group B: topical therapy alone for 4 months, topical therapy + CIT/HOMO for 4 months, 2 months of wash out)	Daily oral intake of the fixed combination CIT/HOMO for 4 months improved the function of inner retinal cells recorded by PERG in the inferior and in the superior quadrants, independently of IOP reduction	[66]
Kim et al. (2021)	Quinoid pigments	Review	Histochrome is characterized by hemoresorption, retinoprotective and antioxidant properties, showing efficacy in various eye pathologies, including retinal hemorrhages of various etiologies, postoperative hyphema, diabetic retinopathy, keratitis, endothelial–epithelial dystrophia of the cornea, postoperative corneal edema, central retinal vein thrombosis and retinal dystrophy	[35]

## Data Availability

The data can be shared up on request.

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
