# Peer review of "Marine Natural Products Rescuing the Eye: A Narrative Review"

_marinedrugs, 2024, doi:10.3390/md22040155_

Round 1

Reviewer 1 Report

Comments and Suggestions for Authors

It is interesting a review which examine the different way of reference #4, about the treatment and/or prevention effect of marine compounds on eye diseases, 

Authors summarized the effects and marine compounds on eye diseases in Tables. I think the Table1, 2 need the references and also needs indication Table number in Text.

For example:

Line 86: (TNF-alpha, VEGF)[13](Table 1).

Line 413: aqueous humor in uveitis [12]

Line 418: cellular infiltration and cytokines [13]

Author Response

It is interesting a review which examine the different way of reference #4, about the treatment and/or prevention effect of marine compounds on eye diseases.

Authors summarized the effects and marine compounds on eye diseases in Tables. I think the Table1, 2 need the references and also needs indication Table number in Text.

For example:

Line 86: (TNF-alpha, VEGF)[13](Table 1).

Line 413: aqueous humor in uveitis [12]

Line 418: cellular infiltration and cytokines [13]

RE: Thank you very much for your comments. We updated the manuscript adding the references in the Tables and adding the Table number in the Text.

Reviewer 2 Report

Comments and Suggestions for Authors

I have read the manuscript and have several questions and comments.
1. Among marine pigments, the so-called quinoid pigments are widely studied. Due to their wide distribution, chemical diversity and scientifically confirmed pharmacological properties, sea urchin pigments have evoked renewed interest as a promising source material for the development of drugs that might be useful in clinical practice for the treatment of various human diseases. They have shown good activity for the treatment of eye diseases. It is necessary to supplement the manuscript with information about pigments.
2. pigments, predominantly contained polyhydroxy-1,4-naphthoquinone, from green sea urchin (Strongylocentrotus droebachiensis) shells were studied on animal models – guinea pig ileum contraction, rabbit eyes allergic conjunctivitis, and rabbit local skin irritation. It is significantly reduced, in a dose-dependent manner, the histamine-induced contractions of the isolated guinea pig ileum with ID50 = 1.2 µg/mL (in equivalents of spinochrome B), had an inhibitory effect on the model of ocular allergic inflammation surpassing the reference drug olopatadine, and did not show any irritating effect in rabbits.
3. Echinochrome A, isolated from sea urchins, is registered as Animal-derived medicinal products in Russia in the treatment of glaucoma, intraocular haemorrhages at different locations and intensities, in degenerative processes, for dystrophic lesions of the cornea and for inflammatory diseases of the eye.
4. Tables 1 and 2 need to be updated. Please include the name of the molecule/extract, information about the source, model, dose/concentration range, presence of positive/negative controls, results in numbers, references.
5. To increase interest in the manuscript, it is necessary to supplement the manuscript with information about clinical studies. Arrange them in the form of a table indicating the molecule, source for isolation, type of study, results. Comment on the data presented.
6. The manuscript is designed chaotically.
7. The list of references is designed very carelessly.
8. The authors chose a rather narrow aspect of the use of marine products, namely for the treatment of eyes. It is necessary to supplement information specifically for the treatment of the eyes, on relevant models, and exclude accompanying information about general data, such as anti-inflammatory, etc. Information should only be, for example, anti-inflammatory, anti-allergic, antioxidant, etc. activity on the eyes.
9. The manuscript requires significant revision in accordance with the stated topic. As presented, I cannot recommend it for publication in Marina Drugs.

Author Response

I have read the manuscript and have several questions and comments.

1. Among marine pigments, the so-called quinoid pigments are widely studied. Due to their wide distribution, chemical diversity and scientifically confirmed pharmacological properties, sea urchin pigments have evoked renewed interest as a promising source material for the development of drugs that might be useful in clinical practice for the treatment of various human diseases. They have shown good activity for the treatment of eye diseases. It is necessary to supplement the manuscript with information about pigments.

RE: Thank you very much for your precious suggestions. We added a paragraph in the Results section called 3.11 Quinoid pigments (page 8, lines 414-443)

3.11. Quinoid Pigments

Quinoid pigments are organic phenolic compounds isolated from the shells and spines of sea urchins. These have been extensively studied for their widely possible utilities and particurly for their medical applications [78]. Moreover, thanks to their anti-oxidant, anti-bacterial, anti-viral, anti-inflammatory and anti-allergic properties, quinoids have been utilized in managing ocular disorders. Pozharitskaya and collegues evaluated the effect of an extract of pigments from green sea urchin Strongylocentrotus droebachiensis containing predominantly polyhydroxy-1,4-naphthoquinone. In rabbit, the extract showed a therapeutic effect, without irritation, on induced allergic conjunctivitis, displaying a more effective action in comparison with olopatadine, the reference drug for ocular allergies [79]. Echinochrome A, one of the most abundant quinonoid pigment in sea urchins, demonstrated to be effective in treating traumatic hemophthalmia in rabbits. Moreover, the administration of Echinochrome A enhanced corneal epithelialization, decreased corneal defect, reduced the risk of corneal perforation and alleviated eye inflammation [80] (Table 1). Thus, Histochrome, a drug preparation containing sodium and echinochrome A, was approved in Russia for the treatment of several eye conditions including glaucoma, intraocular haemorrhages, degenerative retinal and corneal diseases, vascular and inflammatory ocular disorders [81]. Clinical studies demonstrated that Histochrome effectively improved visual acuity and reduced hemorrhage and inflammation [81,82]. Particurarly, it demonstrated efficacy in resolving hyphema, retinal hemorrhages, and diabetic retinopathy. Additionally, Histochrome exhibited retinoprotective properties and enhanced blood flow in the posterior pole of the eye. Positive results were also obtained when keratitis and corneal dystrophy were treated with Histochrome [83]. In addition, in patients with primary open-angle glaucoma (POAG) undergoing trabeculectomy, combining Histochrome with magnetotherapy led to improved visual function and stabilized glaucoma progression during the early postoperative period [81] (Table 2).

2. pigments, predominantly contained polyhydroxy-1,4-naphthoquinone, from green sea urchin (Strongylocentrotus droebachiensis) shells were studied on animal models – guinea pig ileum contraction, rabbit eyes allergic conjunctivitis, and rabbit local skin irritation. It is significantly reduced, in a dose-dependent manner, the histamine-induced contractions of the isolated guinea pig ileum with ID50 = 1.2 µg/mL (in equivalents of spinochrome B), had an inhibitory effect on the model of ocular allergic inflammation surpassing the reference drug olopatadine, and did not show any irritating effect in rabbits.

RE: Thank you very much for your precious suggestions. We added a paragraph in the Results section called 3.11 Quinoid pigments (page 8, lines 414-443)

3.11. Quinoid Pigments

Quinoid pigments are organic phenolic compounds isolated from the shells and spines of sea urchins. These have been extensively studied for their widely possible utilities and particurly for their medical applications [78]. Moreover, thanks to their anti-oxidant, anti-bacterial, anti-viral, anti-inflammatory and anti-allergic properties, quinoids have been utilized in managing ocular disorders. Pozharitskaya and collegues evaluated the effect of an extract of pigments from green sea urchin Strongylocentrotus droebachiensis containing predominantly polyhydroxy-1,4-naphthoquinone. In rabbit, the extract showed a therapeutic effect, without irritation, on induced allergic conjunctivitis, displaying a more effective action in comparison with olopatadine, the reference drug for ocular allergies [79]. Echinochrome A, one of the most abundant quinonoid pigment in sea urchins, demonstrated to be effective in treating traumatic hemophthalmia in rabbits. Moreover, the administration of Echinochrome A enhanced corneal epithelialization, decreased corneal defect, reduced the risk of corneal perforation and alleviated eye inflammation [80] (Table 1). Thus, Histochrome, a drug preparation containing sodium and echinochrome A, was approved in Russia for the treatment of several eye conditions including glaucoma, intraocular haemorrhages, degenerative retinal and corneal diseases, vascular and inflammatory ocular disorders [81]. Clinical studies demonstrated that Histochrome effectively improved visual acuity and reduced hemorrhage and inflammation [81,82]. Particurarly, it demonstrated efficacy in resolving hyphema, retinal hemorrhages, and diabetic retinopathy. Additionally, Histochrome exhibited retinoprotective properties and enhanced blood flow in the posterior pole of the eye. Positive results were also obtained when keratitis and corneal dystrophy were treated with Histochrome [83]. In addition, in patients with primary open-angle glaucoma (POAG) undergoing trabeculectomy, combining Histochrome with magnetotherapy led to improved visual function and stabilized glaucoma progression during the early postoperative period [81] (Table 2).

3. Echinochrome A, isolated from sea urchins, is registered as Animal-derived medicinal products in Russia in the treatment of glaucoma, intraocular haemorrhages at different locations and intensities, in degenerative processes, for dystrophic lesions of the cornea and for inflammatory diseases of the eye.

RE: Thank you very much for your precious suggestions. We added a paragraph in the Results section called 3.11 Quinoid pigments (page 8, lines 414-443)

3.11. Quinoid Pigments

Quinoid pigments are organic phenolic compounds isolated from the shells and spines of sea urchins. These have been extensively studied for their widely possible utilities and particurly for their medical applications [78]. Moreover, thanks to their anti-oxidant, anti-bacterial, anti-viral, anti-inflammatory and anti-allergic properties, quinoids have been utilized in managing ocular disorders. Pozharitskaya and collegues evaluated the effect of an extract of pigments from green sea urchin Strongylocentrotus droebachiensis containing predominantly polyhydroxy-1,4-naphthoquinone. In rabbit, the extract showed a therapeutic effect, without irritation, on induced allergic conjunctivitis, displaying a more effective action in comparison with olopatadine, the reference drug for ocular allergies [79]. Echinochrome A, one of the most abundant quinonoid pigment in sea urchins, demonstrated to be effective in treating traumatic hemophthalmia in rabbits. Moreover, the administration of Echinochrome A enhanced corneal epithelialization, decreased corneal defect, reduced the risk of corneal perforation and alleviated eye inflammation [80] (Table 1). Thus, Histochrome, a drug preparation containing sodium and echinochrome A, was approved in Russia for the treatment of several eye conditions including glaucoma, intraocular haemorrhages, degenerative retinal and corneal diseases, vascular and inflammatory ocular disorders [81]. Clinical studies demonstrated that Histochrome effectively improved visual acuity and reduced hemorrhage and inflammation [81,82]. Particurarly, it demonstrated efficacy in resolving hyphema, retinal hemorrhages, and diabetic retinopathy. Additionally, Histochrome exhibited retinoprotective properties and enhanced blood flow in the posterior pole of the eye. Positive results were also obtained when keratitis and corneal dystrophy were treated with Histochrome [83]. In addition, in patients with primary open-angle glaucoma (POAG) undergoing trabeculectomy, combining Histochrome with magnetotherapy led to improved visual function and stabilized glaucoma progression during the early postoperative period [81] (Table 2).

4. Tables 1 and 2 need to be updated. Please include the name of the molecule/extract, information about the source, model, dose/concentration range, presence of positive/negative controls, results in numbers, references.

RE: Thank you very much for your comments. We updated the Tables 1 and 2 as you suggested (pages 10-13)

5. To increase interest in the manuscript, it is necessary to supplement the manuscript with information about clinical studies. Arrange them in the form of a table indicating the molecule, source for isolation, type of study, results. Comment on the data presented.

RE: Thank you very much for your suggestion. We added a Table (Table 3) in the Manuscript (pages 14-18), trying to supplement the manuscript with information about clinical studies.

6. The manuscript is designed chaotically.

RE: Thank you very much for your comment. We are sorry for the mistake, we revised all the manuscript and we moved the materials and methods section after Introduction section. Then, in the Results section, we tried to summarize all the main natural marine molecules with scientific data published in the literature which have potential beneficial effects on the eyes. We also added another Table to the manuscript (Table 3), as you suggested.

7. The list of references is designed very carelessly.

RE: Thank you very much for your suggestion and we are sorry for the mistake. We revised all the references list in the Manuscript and in the text.

8. The authors chose a rather narrow aspect of the use of marine products, namely for the treatment of eyes. It is necessary to supplement information specifically for the treatment of the eyes, on relevant models, and exclude accompanying information about general data, such as anti-inflammatory, etc. Information should only be, for example, anti-inflammatory, anti-allergic, antioxidant, etc. activity on the eyes.

RE: Thank you very much for your comments. For each natural marine molecule listed in our review we tried to show their healing properties and then we focused on these beneficial effects for the eye, also trying to be more specific with the help of Tables 1 and 2, dividing the therapeutic effects on ocular anterior segment (Table 1) and ocular posterior segment (Table 2), also trying to avoid possible repetitions between Main Text and Tables.

9. The manuscript requires significant revision in accordance with the stated topic. As presented, I cannot recommend it for publication in Marina Drugs.

RE: Thank you very much for your comment. We tried to improve the manuscript according to all your suggestions. We hope that now the manuscript is suitable for publication in Marine Drugs.